# The Use of PuraStat^®^ in the Management of Walled-Off Pancreatic Necrosis Drained Using Lumen-Apposing Metal Stents: A Case Series

**DOI:** 10.3390/medicina59040750

**Published:** 2023-04-12

**Authors:** Cecilia Binda, Alessandro Fugazza, Stefano Fabbri, Chiara Coluccio, Alessandro Repici, Ilaria Tarantino, Andrea Anderloni, Carlo Fabbri

**Affiliations:** 1AUSL Romagna, Gastroenterology and Digestive Endoscopy Unit, Forlì-Cesena Hospitals, 47121 Forlì-Cesena, Italy; 2Humanitas Research Hospital, Digestive Endoscopy Unit, Division of Gastroenterology, Rozzano, 20089 Milan, Italy; 3Humanitas University, Department of Biomedical Sciences, 20090 Pieve Emanuele, Italy; 4Digestive Endoscopy and Gastroenterology Unit, Department of Gastroenterology, Istituto Mediterraneo per i Trapianti e Terapie ad Alta Specializzazione (IsMeTT/UPMC), 90127 Palermo, Italy; 5Fondazione I.R.C.C.S. Policlinico San Matteo, Gastroenterology and Digestive Endoscopy Unit, 27100 Pavia, Italy

**Keywords:** PuraStat, walled-off pancreatic necrosis, pancreatic fluid collection, LAMS, EUS-guided drainage, bleeding, endoscopic hemostasis, endoscopic hemostatic agents

## Abstract

*Background and Objectives*: Bleeding is one of the most feared and frequent adverse events in the case of EUS-guided drainage of WOPN using lumen-apposing metal stents (LAMSs) and of direct endoscopic necrosectomy (DEN). When it occurs, its management is still controversial. In the last few years, PuraStat, a novel hemostatic peptide gel has been introduced, expanding the toolbox of the endoscopic hemostatic agents. The aim of this case series was to evaluate the safety and efficacy of PuraStat in preventing and controlling bleeding of WOPN drainage using LAMSs. *Materials and Methods*: This is a multicenter, retrospective pilot study from three high-volume centers in Italy, including all consecutive patients treated with the novel hemostatic peptide gel after LAMSs placement for the drainage of symptomatic WOPN between 2019 and 2022. *Results*: A total of 10 patients were included. All patients underwent at least one session of DEN. Technical success of PuraStat was achieved in 100% of patients. In seven cases PuraStat was placed for post-DEN bleeding prevention, with one patient experiencing bleeding after DEN. In three cases, on the other hand, PuraStat was placed to manage active bleeding: two cases of oozing were successfully controlled with gel application, and a massive spurting from a retroperitoneal vessel required subsequent angiography. No re-bleeding occurred. No PuraStat-related adverse events were reported. *Conclusions*: This novel peptide gel could represent a promising hemostatic device, both in preventing and managing active bleeding after EUS-guided drainage of WON. Further prospective studies are needed to confirm its efficacy.

## 1. Introduction

Walled-off necrosis (WON) is one of the possible local complications of acute necrotizing pancreatitis (ANP) that could be life-threatening when it becomes symptomatic (e.g., signs of systemic infections, gastric or intestinal outflow obstruction, abdominal pain, compression on major vessels, jaundice) [1]. The management of these collections has been largely studied in the last years, and the paradigm of treatment has changed over the years. The current international guidelines suggest adopting the so called “step-up approach”, indicating a gradual increase from a less invasive procedure to a more invasive one [2,3]. The endoscopic ultrasound (EUS)-guided drainage followed by direct endoscopic necrosectomy (DEN) nowadays represents the treatment of choice in the case of symptomatic WON, especially after the introduction on the market of dedicated devices, such as lumen-apposing metal stents (LAMSs) [2]. These stents have been demonstrated to be very effective for the drainage of WON but are burdened by non-negligible rates of adverse events (AEs), up to 20% [4]. 

Bleeding is one of the most common and feared AEs in the case of EUS-guided drainage of WON using LAMSs and of DEN, and when it occurs, it usually requires urgent interventions, such as endoscopy or radiological embolization or coiling. A large international multicenter series published by Fugazza et al. [5] reported bleeding in 27.8% of all LAMS-related AEs in the setting of pancreatic fluid collections (PFCs), with an overall risk of 7.2% (22/304). In a retrospective analysis involving 30 Italian centers over a 5-year period, including 269 WON, bleeding was reported as the most frequent AEs, occurred in 6% of the patients [6]. When it occurs, the management of bleeding is still a dilemma, mainly because of the difficulties in obtaining effective hemostasis and because of the “extraluminal”/peritoneal location. 

In the last few years, a novel hemostatic peptide gel has been introduced, expanding the toolbox of the endoscopic hemostatic agents. PuraStat^®^ (3-D Matrix Europe SAS, Caluire et Cuire, France) is a synthetic hemostatic agent licensed as CE and marketed as a surgical hemostatic agent. PuraStat is a liquid that when applied to a bleeding area acts rapidly to form a gel coat, which induces hemostasis. It is a slightly viscous solution of synthetic peptides. Contact between PuraStat and blood causes the acidic peptide solution to be neutralized and exposed to ions, resulting in the formation of ß-sheets that then form a three-dimensional scaffold structure. For this reason, it has both a role for the treatment of active bleeding and the potential of enhancing endoscopic mucosal wound healing, preventing delayed bleeding. 

PuraStat is applied through a small catheter placed through the biopsy channel of the endoscope and can be used in various small spaces. It is supplied in a pre-filled syringe and is currently available in 1 mL, 3 mL, and 5 mL unit doses indicated for hemostasis in several surgical circumstances. On application to tissues, PuraStat forms a fully transparent, slightly viscous aqueous peptide (2.5%) solution over the bleeding or potentially bleeding area. PuraStat has several unique features that distinguish it from existing products. As advantages, it is an inert material with no risk of contamination from a biological source, it is available in single prefilled and ready-to-use syringes, requiring no preparation, and can be used repetitively. The transparent adherent barrier permits further endoscopic therapy to be performed, and it can be applied in the general area of bleeding and does not require precise application on the exact point of bleeding. Moreover, it can be removed easily if desired, and it is applicable to narrow spaces.

For all these reasons, PuraStat could represent a promising hemostatic device for many endoscopic procedures, including EUS-PFCs drainage. Therefore, the aim of this study was to evaluate the safety and efficacy of this new hemostatic agent in preventing and controlling bleeding after LAMSs drainage in the setting of WON.

## 2. Materials and Methods

The present case series is a multicenter, retrospective pilot study of a prospectively maintained database from three tertiary Italian institutions, including all consecutive patients treated with PuraStat after LAMSs placement for the drainage of symptomatic WON between June 2019 and February 2022. 

Mature pancreatic WON was defined as encapsulated collection containing necrotic debris, as defined by the Atlanta classification [1]. Indications for drainage of WON included infections and symptoms of obstruction and abdominal pain attributable to WON.

Baseline characteristics, including age, sex, etiology of pancreatitis, location, and size of WON, percentage of estimated necrosis, and indication to WON intervention, were collected. Preprocedural bleeding risk factors, such as antithrombotic or anticoagulant therapy or presence of a vessel within the WON on cross-sectional imaging, were identified. 

### 2.1. Procedure

All patients signed adequate informed consent prior to any interventional procedure [7]. All the initial drainage procedures were performed by experienced endoscopists with a linear array echoendoscope under general anesthesia. Under EUS and fluoroscopic guidance, the WON was drained from either the stomach or duodenum with the deployment of an electrocautery LAMS (Hot-AXIOS; Boston Scientific Corp, Marlborough, MA, USA) (Figure 1). The optimal puncture site, avoiding vasculature, was confirmed under EUS and Doppler flow guidance. All LAMSs were released using the free-hand technique, with direct access into the collection by puncture with the electrocautery system and without a guidewire. The second flange of the stent was released with the intra-channel stent release technique [8]. The 15 × 10 or 20 × 10 mm LAMSs were chosen according to the size of the collection with the amount of necrosis at the discretion of the endoscopist. Pneumatic dilation of the LAMSs, hydrogen peroxide irrigation of the cavity, placement of double-pigtail stent through the LAMSs, and/or immediate extraction of necrotic debris at the time of WON drainage were left at the discretion of the endoscopist. All patients were under broad-spectrum antibiotic therapy at the moment of LAMS placement.

Then, DEN was performed through LAMS with a forward-viewing endoscope at regular intervals. Endoscopic necrosectomy was performed with a combination of sucking debris through the working channel, removing necrotic material with a removal device, and applying irrigation. Endoscopists could use various accessories for fragmentation and removal of necrotic debris, including conventional cold snares, baskets, roth nets, or novel dedicated devices, such as the EndoRotor Powered Endoscopic Debridement (PED) System^®^ (Interscope Medical, Inc., Worcester, MA, USA) or Necrolit^®^ (Meditalia s.a.s., Palermo, Italy). Finally, debris after copious lavage were aspirated, while hydrogen peroxide irrigation of the cavity was left at the discretion of the endoscopists. DEN sessions were performed until complete debridement of necrotic material was achieved and granulation of pink tissue in almost all walls of the cavity was seen.

PuraStat was then applied both for the treatment of active intraprocedural bleeding and for the prevention of delayed bleeding at the end of the DEN sessions (Figure 1). PuraStat was placed at the bleeding source, alone or in combination with other conventional hemostatic techniques, for the control of intraprocedural active bleeding. Instead, after the DEN sessions, the hemostatic gel was placed inside the WON cavity on the newly exposed granulation tissue after removal of necrotic debris. Thus, PuraStat was applied on the most vascularized areas of granulation tissue to prevent delayed bleeding from microscopic vessels and to promote cystic wall healing.

### 2.2. Data Collections and Analysis

Intraprocedural bleeding was defined as visible bleeding within the WON cavity or on the gastric wall, identified during the initial drainage procedure or subsequent DENs. Delayed bleeding was defined as the occurrence of clinical signs of bleeding (hematemesis, melena, or a drop in hemoglobin >2 g/dL) and the presence of fresh blood or stigmata of recent bleeding on endoscopic investigation after the procedure when PuraStat was placed to prevent a primary bleeding event. Rebleeding was defined as secondary bleeding event when PuraStat was used alone or in combination with other hemostatic techniques in the management of the primary active bleeding. Technical success of PuraStat application was defined by the ability of gel deployment in the target area. Posology, time for application, and PuraStat-related AEs were recorded. Additional endoscopic or radiological hemostatic techniques were decided by the endoscopist according to severity of the hemorrhage.

The primary endpoints were to assess the clinical success of PuraStat, meaning its ability to achieve immediate hemostasis in the treatment of active bleeding and to prevent delayed bleeding and the rate of PuraStat-related AEs. The secondary outcomes assessed were the severity of bleeding and type and number of additional interventions needed for the management of bleeding in the setting of WON. 

Technical success of EUS-drainage was defined as placement of the LAMSs for PFC-drainage. Necrotic tissue was debrided during one or more DEN session until complete resolution. Clinical success of DEN was established as WON < 2 cm on cross-sectional imaging and symptoms resolution without need for further interventional radiologic or surgical procedures. All LAMSs were removed within 4 weeks after placement, and the patients were followed up for at least 30 days after LAMS removal.

All AEs related to LAMS placement and DEN procedure, including bleeding, were classified according to the ASGE Lexicon severity grading system [9], and time of occurrence and their management were reported. 

## 3. Results

### 3.1. Baseline Characteristics

During the study period, a total of ten patients (nine men, one woman) were enrolled (Table 1), with a mean age of 65.7 years (range 24–89 years). The etiologies of acute pancreatitis were 70% gallstones, 20% idiopathic, and 10% alcohol. Indications for EUS-WON were abdominal pain in six patients (60%), infected WON in six (60%), and gastric outlet obstruction in five (50%), and early satiety in two (20%). At the onset of pancreatitis, two patients were on anticoagulant therapy with Rivaroxaban, which was shifted to low molecular weight heparin (LMWH) during their hospital stays; according to the recent guidelines [10], LMWH was omitted the day of the procedure. Before drainage, all the patients underwent contrast-enhanced abdominal CT-scan to assess WON characteristics. WON was located in the head of the pancreas in two patients (20%), in the body-tail in five (50%), while it affected the entire pancreatic gland in three cases (30%). The medium WON diameter was 126.8 mm (range 82–180 mm). Pancreatic necrosis was estimated to be greater than 50% in four patients (40%). Three patients had vessels within the WON seen on cross-sectional imaging before LAMS placement; after a multidisciplinary discussion with interventional radiologists and surgeons, prophylactic coil embolization of the splenic artery was pursued in one case, while in another patient the DEN procedure was performed in an angiographic room without the need of arterial embolization. The third patient did not require any radiological treatment; thus, conventional EUS-guided drainage and subsequent DEN were performed in the endoscopic room. None of these patients experienced bleeding events during the study period.

### 3.2. Procedure Details

Details of EUS-drainage, DEN and PuraStat application are shown in Table 2. Technical success of WON drainage using LAMSs was 100%. The 20 × 10 mm LAMSs were used in nine cases, while 15 × 10 mm LAMSs were deployed during three initial drainage procedures. Two patients (20%) were treated with two LAMSs with a multiple gateway drainage strategy, as previously reported [11]. In all cases, LAMSs were deployed through the gastric wall. Pneumatic dilation of the LAMSs were performed in nine patients (90%), while the WON cavity was irrigated with hydrogen peroxide in seven patients (70%). 

DEN was subsequently performed using conventional accessories, including snares and Dormia baskets, respectively, in ten and three cases. In addition, two patients were treated with novel dedicated necrosectomy devices, such as EndoRotor^®^ and Necrolit^®^. Clinical success of DEN was obtained in 100% of the patients. The mean number of DEN sessions needed to achieve the complete debridement of the WON was 1.6 (range 1–3), with four patients that required more than one DEN session. Double pig-tail plastic stents were placed through the LAMSs at the end of the DEN procedures in five cases (50%). Technical success of PuraStat application was achieved in 100% of patients in the mean time of 4 min (range 2–6 min). The mean posology of PuraStat applied after each DEN session was 3 mL. In two cases, the novel hemostatic gel was used during two sessions.

### 3.3. Bleeding Management and Prevention with PuraStat

Outcomes of PuraStat application in the management and prevention of WON-related bleeding are reported in Table 3. In three patients out of ten, PuraStat was used in the management of WON-related active bleeding. None of these patients were previously treated with PuraStat for prophylactic purpose. In one patient, blood oozing from the gastric wall was seen during the drainage procedure immediately after LAMS deployment and was successfully treated with PuraStat application after multiple unsuccessful attempts with Trough-The-Scope Clips (TTSC). The second patient experienced blood oozing 12 h after a DEN session from a minor WON vessel, which was effectively treated with PuraStat without the need for additional conventional hemostatic techniques. The third patient had massive spurting bleeding from a large retroperitoneal vessel during a DEN session. Endoscopic hemostasis with clips was initially attempted, but the source of bleeding could not be for certain identified because of the large amount of blood clots within the collection. As hemodynamic instability succeeded, PuraStat was applied inside the cavity as a bridge hemostatic treatment to the subsequent radiological embolization. Indeed, after fluid resuscitation and blood transfusions, the patient was referred to the angiographic room, but as no active bleeding was identified, embolization was not needed. The patient was admitted to the intensive care unit, and no further hemorrhages occurred. No case of re-bleeding occurred.

In the other seven cases, PuraStat was placed on the WON walls to prevent post-DEN bleeding. Of these seven patients, only one experienced delayed bleeding 36 h after DEN. After blood transfusion, the patients underwent endoscopy examinations, which did not show any active bleeding, without the need for additional hemostatic techniques. No PuraStat-related AEs occurred.

However, two different procedure-related complications were also reported: a severe periprocedural respiratory failure requiring ICU admission and a mild pneumonia conservatively treated.

## 4. Discussion

The endoscopic management of intracystic bleeding is still a dilemma for the endoscopist because of the difficulties of obtaining adequate hemostasis with the available devices and the lack of dedicated ones. Several endoscopic hemostatic techniques for the management of PFC-related hemorrhage has been reported: epinephrine injection, hemoclip placement, coagulation [12], balloon tamponade [13], fibrin glue injection [14], hemostatic powder application [15], and covered SEMS placement [16,17]. However, there is no consensus on the optimal endoscopic hemostasis. 

The use of PuraStat in gastrointestinal endoscopy is approved for the management of bleeding from small vessels and oozing from capillaries of the GI tract and the surrounding tissues. Moreover, PuraStat application showed a reduction of delayed bleeding following gastrointestinal endoscopic submucosal dissection (ESD) procedures in the colon [18,19].

In a recently published prospective multicenter study, Branchi et al. [20] investigated the role of PuraStat in providing hemostasis in 111 patients with acute non-variceal gastrointestinal bleeding. When PuraStat was used as the primary therapy, initial hemostatic success was reached in 94% of patients (74/79, 95% CI 88–99%). PuraStat induced hemostasis in 75% of patients (24/32, 95% CI 59–91%) if used as a secondary treatment option after failure of standard techniques. The volume of gel required to achieve hemostasis was 3 mL in 59% of cases. Overall, the rebleeding rate at the 30-day follow-ups was 16% (18/111). No adverse events due to application of PuraStat or technical failures were reported.

Soriani et al. [21] reported a successfully PuraStat-managed bleeding from intrahepatic biliary ducts that occurred during cholangioscopy-guided lithotripsy. This case report confirmed the opportunity of PuraStat placement also in bleeding sites that are inaccessible using conventional endoscopic hemostatic devices as it could happen inside the necrotic cavities during DEN. 

First, in our case series, PuraStat was used in the management of three cases of WON-related active bleeding. The two blood-oozing events, one from the gastric wall and one from inside the cavity, were successfully controlled with PuraStat application. In one case of massive blood spurting causing hemorrhagic shock, PuraStat was used as a bridge hemostatic agent to temporarily control the blood loss, allowing the patient to be referred to the angiographic room after hemodynamic resuscitation. These data are in line with the results of previous studies, which demonstrated that PuraStat is much more effective in achieving initial hemostasis in cases of oozing than for spurting bleeding. In fact, Subramaniam et al. [19] reported that PuraStat could provide hemostasis in 72.6% of oozing bleeding cases but only in 50.0% of spurting bleeding cases in gastrointestinal tract. Moreover, PuraStat can be used as a rescue hemostatic agent to achieve temporary hemostasis of massive arterial bleeding. Likewise to what was reported in our cohort, Branchi et al. [20] described five cases in which PuraStat was used a bridge to surgery in the case of refractory severe bleeding. 

The use of PuraStat in the treatment of acute bleeding inside a pseudocyst after EUS-guided drainage using LAMS was anecdotally described for the first time by De Nucci et al. [22]. In their cohort, PuraStat was applied into the pseudocyst cavity as the primary hemostatic agent, and hemorrhage was controlled after the application of 6 mL of gel. After 3 days, a second-look endoscopy revealed granulation tissue on the cyst cavity. No rebleeding occurred, and the LAMS was removed 2 weeks later. 

Secondarily, this is the first pilot study to investigate the role of PuraStat in the prevention of WON-related delayed bleeding. In light of the reduction of delayed bleeding following endoscopic resections in the gastrointestinal tract [18], we applied PuraStat on the WON walls as a prophylactic hemostatic agent in seven cases, and only one patient experienced moderate delayed bleeding with no need for subsequent additional endoscopic hemostasis. Heretofore, a technique described for the prevention of bleeding after transmural drainage of PFC was the co-axial DPPSs placement within the LAMS lumen. It has been hypothesized that the placement of the co-axial DPPS through the LAMS could have a protective effect in preventing the impaction and the friction of the distal flange against the adjacent wall of the cavity, which would reduce the risk of bleeding [8]. The role of DPPS in preventing bleeding after LAMS drainage is still matter of debate. The first single center study by Puga et al. [23] demonstrated that adjunctive placement of DPPS resulted in decreased adverse events, particularly bleeding, while a subsequent large multicenter study reported that deployment of pigtail stents across the LAMSs did not significantly reduce overall adverse events (26% with DPPS (Group 1) versus 27% without DPPS (Group 2); *p* = 0.88.) [24]. 

The rationale for the use of a hemostatic gel, both in prophylaxis and as treatment of active bleeding, comes from the idea that WON bleeding has several mechanisms. Intra-procedural bleeding, indeed, can originate at the puncture site from small missed venous collaterals of the gastric or duodenal wall, from minor vessels within the cavity, or from the large retroperitoneal vessel that may bleed after rapid collapse of the collection or due to iatrogenic damage during necrosectomy [25]. In this setting, the hemostatic gel can therefore be useful both to prevent bleeding, especially when originating from minor vessels within the cavity, or to treat active bleeding of the gastric or duodenal wall or from the vessel of the cystic wall. 

Instead, delayed bleeding is usually caused by prolonged contact between the distal flange of the indwelling LAMS and the walls that can lead to the erosion of intracavitary vessels and promote the formation of a pseudoaneurysm [25]. Thus, LAMS removal within 4 weeks of deployment is recommended, especially if imaging shows resolution of the cavity [26,27]. In the case of pseudoaneurysm, the role of Purastat may be marginal, although it may be useful as a bridge to other definitive treatments, namely radiological ones.

Jiang et al. [28] proposed an algorithm for the management of WON-related bleeding based on the stratification of bleeding severity (adapted in Figure 2). Mild bleeding occurring either during initial drainage or during necrosectomy or caused by stent erosion after the procedure could be treated endoscopically with interventional radiology (IR) or surgery as a backup option. Intra- or post-procedural bleeding believed to come from a pseudoaneurysm should be directly managed by IR embolization [28].

Moreover, in the presence of pseudoaneurysms [29] or arterial vessels [30,31] inside the WON, prophylactic coil embolization could be performed to prevent massive arterial bleeding during drainage or subsequent necrosectomy. Indeed, perigastric varices (OR 2.90, 1.31–6.42, *p* = 0.008) or pseudoaneurysm (OR 2.99, 1.75–11.93, *p* = 0.002) have been identified as independent predictors of overall adverse event occurrences in a large multicenter series [32], while the endoscopic identification of a vessel within the cavity has emerged as a strong predictor of bleeding on multivariate analysis (OR 23.3, 4.0–135.1, *p* < 0.01) in a recent retrospective study by Holmes et al. [33]. In our cohort, prophylactic coil embolization was performed in one of three patients who had evidence of major arterial vessels inside the WON. 

In addition to the known hemostatic effect, it has been hypothesized that the Purastat 3-D structure favors the tissue proliferative process during healing, thanks to its similarity with the natural extracellular matrix. Indeed, PuraStat action in promoting mucosal regeneration has been reported in some recent studies [34,35]. Based on this hypothetical re-epithelizing property, PuraStat application on the walls of the cavity could promote resurfacing, enhancing WON resolution. 

Our case series does have limitations, mainly owing to the small cohort of patients enrolled, its retrospective design, and the lack of a control group; thus, firm conclusions cannot be drawn. Therefore, the potential role of PuraStat in managing and preventing WON-related bleeding and in improving clinical success during necrosectomy must be validated in larger prospective and comparative studies.

## 5. Conclusions

The promising results of our case series confirm that PuraStat expanded the toolbox of the endoscopic hemostatic agents, even in the still debated management of WON-related bleeding. This novel peptide gel is particularly effective in achieving hemostasis for oozing-type bleeding but also plays a role in the management of spurting-type bleeding to temporarily control the blood loss as a bridge hemostatic agent to subsequent radiological or surgical treatments. Moreover, PuraStat can be easily and safely applied for the prevention of delayed post-DEN bleeding and might promote cavity resurfacing, enhancing WON resolution, although these hypotheses require confirmation with ad hoc designed trials.

## Figures and Tables

**Figure 1 medicina-59-00750-f001:**
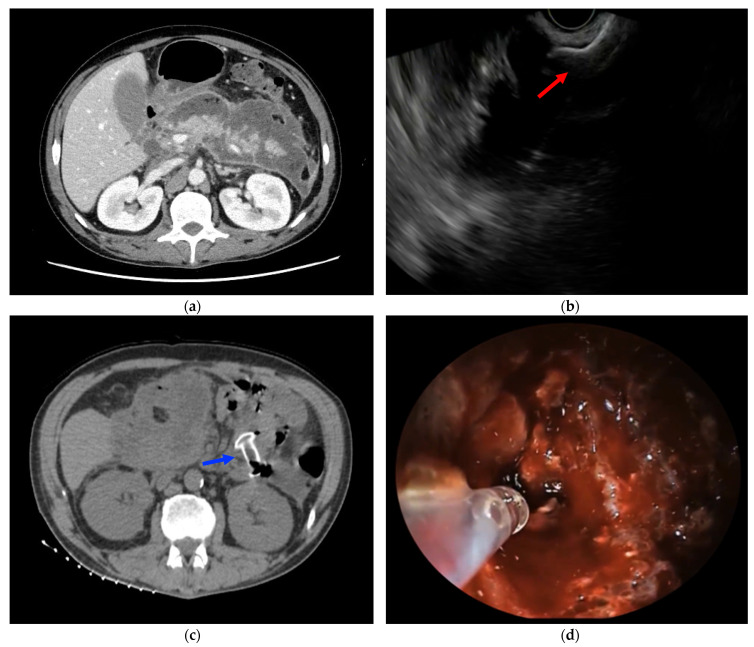
(**a**) CT scan showing a large WON, involving the entire pancreatic gland; (**b**) EUS-WON drainage using LAMS (red arrow); (**c**) CT scan showing LAMS placed between the gastric cavity and the WON (blue arrow); (**d**) PuraStat application on WON walls to treat oozing bleeding from a small intracavitary vessel during DEN.

**Figure 2 medicina-59-00750-f002:**
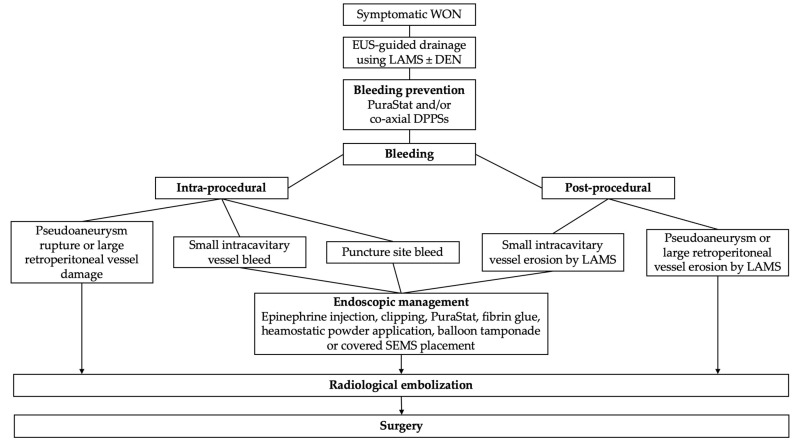
Algorithm for management of WON-related bleeding adapted from Jiang et al. [28] and Rana et al. [25].

**Table 1 medicina-59-00750-t001:** Baseline Characteristics.

Patients’ Characteristics	10 Patients
Gender	
M	9
F	1
Mean Age	65.7 years (24–89)
Etiology of acute pancreatitis	
Gallstones	7
Idiopathic	2
Alcohol	1
Indication for drainage	
Abdominal pain	6
Infected WON	6
Gastric outlet obstruction	5
Early satiety	2
**WON Characteristics**	**10 WON**
WON location	
Body and Tail	5
Entire Pancreas	3
Head	2
Max WON diameter	126.8 mm (82–180)
% of estimated necrosis	
<50% of necrosis	6
>50% of necrosis	4
Evidence of vessel inside WON	3
Prophylactic Embolization	1
DEN performed in angiographic room	1
No need for radiological treatment	1

**Table 2 medicina-59-00750-t002:** Procedure Details.

EUS-Guided WON Drainage	10 Patients
Technical success	100%
LAMS size	
20 × 10 mm	9
15 × 10 mm	3
Multiple Gateway Drainage	2
Trans-Gastric Drainage	10
LAMS dilation	9
Hydrogen peroxide irrigation	7
**Direct Endoscopic Necrosectomy**	**10 Patients**
Devices used for DEN	
Snares	10
Dormia baskets	3
EndoRotor^®^	1
Necrolit^®^	1
Mean DEN sessions	1.6 (1–3)
Clinical success of DEN	10/10
**PuraStat Application**	
Technical success	100%
Mean application time	4 min (2–6)
Mean volume	3 mL

**Table 3 medicina-59-00750-t003:** Bleeding Management and Prevention with PuraStat.

Outcomes	Patients
PuraStat in management of active WON-related bleeding	3
Oozing bleedings successfully treated	2
Bridge to embolization in spurting bleeding	1
PuraStat in post-DEN bleeding prevention	7
Delayed Bleeding	1
PuraStat-related adverse events	0/10

## Data Availability

All articles cited in this article are listed in PubMed.

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
