# Peer review of "The Use of PuraStat® in the Management of Walled-Off Pancreatic Necrosis Drained Using Lumen-Apposing Metal Stents: A Case Series"

_medicina, 2023, doi:10.3390/medicina59040750_

Round 1

Reviewer 1 Report

Thank you for this well written paper on the endoscopic management of pancreatic pseudocysts and potential prevention of bleeding using a new hemostatic agent. .

Few considerations: If you use abbreviations you should put the extended version the first time you use the abbreviation even if this is in the title.  You should add the extended version of LAMS.

The paper is not clear when talking about the number of patients treated and how those were selected, the indication was to treat active bleeding and prophylactic application but apparently three patients were not treated with PuraStat for bleeding prevention.

The authors report one bleeding requiring transfusions at 36 hours and one requiring a second endoscopy at 12 hours, but they do not report those two occurrences as postprocedural bleeding. This is confusing.

If those number are correct there are 2 patients with a postop bleeding out of 7 treated with the new compound, this would represent a percentage of bleeding that seems superior to the number reported in the literature.

There is another case where the compound was used together with a clip and it seems that if a clip was needed the Purastat alone was not sufficient, one more case required embolization and also in this case PuraStat was not successful. The authors again report 100% technical success of PuraStat application with is probably referring to the ability to deploy the compound but this is confusing and they should refer to the ability to control bleeding.

Reviewer 2 Report

Comments are included in the attached file. See also.

Round 2

Reviewer 2 Report

It is well corrected according to the comments, but if you want to correct some of them, you can accept it.

Author Response

Dear reviewer, thank you for your comments.

We changed the tables and subtitle, as you suggested.